# ScienceIoT: Evolution of the Wireless Infrastructure of KREONET

**DOI:** 10.3390/s21175852

**Published:** 2021-08-30

**Authors:** Cheonyong Kim, Joobum Kim, Ki-Hyeon Kim, Sang-Kwon Lee, Kiwook Kim, Syed Asif Raza Shah, Young-Hoon Goo

**Affiliations:** 1Advanced KREONET Center, KISTI, Daejeon 34141, Korea; cykim0807@kisti.re.kr (C.K.); kkh1258@kisti.re.kr (K.-H.K.); sglee@kisti.re.kr (S.-K.L.); wowook@kisti.re.kr (K.K.); 2Department of Information Technology, Middle Georgia State University, Macon, GA 31206, USA; joobum.kim@mga.edu; 3Department of Computer Science, Sukkur IBA University, Airport Road, Delhi Muslim Housing Society, Sukkur 65200, Sindh, Pakistan; asif.shah@iba-suk.edu.pk

**Keywords:** ScienceLoRa, ScienceIoT, KREONET, Internet of Things, low-power wide-area network (LPWAN), LoRa/LoRaWAN, private 5G, edge computing

## Abstract

Here, we introduce the current stage and future directions of the wireless infrastructure of the Korea Research Environment Open NETwork (KREONET), a representative national research and education network in Korea. In 2018, ScienceLoRa, a pioneering wireless network infrastructure for scientific applications based on low-power wide-area network technology, was launched. Existing in-service applications in monitoring regions, research facilities, and universities prove the effectiveness of using wireless infrastructure in scientific areas. Furthermore, to support the more stringent requirements of various scientific scenarios, ScienceLoRa is evolving toward ScienceIoT by employing high-performance wireless technology and distributed computing capability. Specifically, by accommodating a private 5G network and an integrated edge computing platform, ScienceIoT is expected to support cutting-edge scientific applications requiring high-throughput and distributed data processing.

## 1. Introduction

The Internet of Things (IoT) is an innovative networking paradigm suitable for enabling new types of smart applications by interconnecting smart things in various areas, including industrial and public areas as well as academic and scientific fields. With advances in technologies such as wireless communication, smart sensors, and miniaturization, as well as the integration of electronic devices, the IoT infrastructure enables the collection of data from distributed sensors, control over remote actuators, and the interconnection of smart devices. Moreover, the availability of open hardware platforms makes it easier for developers and researchers to create new types of IoT devices for experimenting, testing, demonstrating, and finally deploying new applications. Owing to the proliferation of the IoT, the number of IoT devices has increased significantly. According to the Cisco Annual Internet Report [1], the number of connected devices increases constantly. In particular, machine-to-machine (M2M) applications such as smart metering and healthcare monitoring contribute considerably toward this increase in the number of devices and connections, with an approximately 2.4-fold increase between 2018 and 2023, as shown in Figure 1.

National research and education networks (NRENs) are attempting to build IoT infrastructures in order to improve scientific research methodologies and to enhance the efficiency of public education. Particularly, the transition of the research paradigm from theoretical and experimental research to data-oriented research is associated with the massive data collection ability afforded by the IoT, which requires the deployment of IoT infrastructure for bridging distributed research data and data analysis platforms. Therefore, global NRENs have already recognized the importance of IoT research and are implementing plans to establish IoT infrastructure and services. The Energy Sciences Network (ESNet), a research network of the United States Department of Energy, held a 5G-enabled Energy Innovation workshop for analyzing potential opportunities and technical challenges in adopting wireless technologies in research fields. SURFnet, a research network of the Netherlands, has managed the Unwired Campus Working Group for integrating wired infrastructure and wireless technologies, such as private 5G and wireless sensor networks. The Science Information Network (SINET), a Japanese academic backbone network, has built a wireless infrastructure based on mobile networks in order to collect research data for application in areas such as agriculture, forestry, medical science, and social science.

This article presents the concept, current status, applications, and development plans of the wireless infrastructure for science and education in the Korea Research Environment Open NETwork (KREONET). In addition to the worldwide efforts for building wireless infrastructure for research, KREONET has launched ScienceLoRa, a wireless IoT infrastructure based on a low-power wide-area network (LPWAN) technology [2]. KREONET has a powerful wired backbone network, the appropriate software stack for ensuring high-performance data transmission, and interactions with supercomputing infrastructure. Based on its wired capacity, the main challenge faced by ScienceLoRa is the integration of wired and wireless network environments to transmit, process, and analyze research data obtained from wireless IoT devices. Currently, ScienceLoRa provides various scientific applications such as environmental monitoring, research facility management, and research data collection. Since 2018, 25 ScienceLoRa sites have been deployed across Korea, and over 50 research devices have been placed in operation to generate scientific data through the ScienceLoRa network. Currently, ScienceLoRa is evolving toward ScienceIoT to expand its service coverage and improve service quality.

Although LPWAN technology enables building a low-rate and long-range wireless network efficiently, there are two main concerns when it is used for scientific research networks. First, the data rate of LPWAN is not enough for delivering multimedia data which are actively used in data-oriented research projects. Second, the simple network architecture (i.e., star topology) will bring network congestion when the number of connected devices increases. This data concentration would degrade the quality and availability of ScienceLoRa services. To overcome these limitations, we are developing ScienceIoT, which incorporates various wireless technologies and edge computing. ScienceIoT will be able to support a variety of scientific research fields requiring high-performance wireless connections. Moreover, the quality of the service will be enhanced, and the load on the core network will be decreased by processing data at the edge.

The remainder of this article is organized as follows. Section 2 provides background information regarding IoT wireless technologies and KREONET. Section 3 presents the current state of ScienceLoRa, including the system architecture, hardware platform, and network and communication architectures. The current services of ScienceLoRa are discussed in Section 4, while Section 5 explains ScienceIoT and the future directions. Lastly, Section 6 presents the conclusions of this work.

## 2. Background

The necessity of building wireless infrastructure in research fields has been actively indicated by NRENs such as ESNet, SURFnet, and SINET. They have raised several points regarding research activities conducted without wireless infrastructure. First, last-mile connectivity should be provided to researchers, experimental equipment, and research facilities. In this context, wireless infrastructure built for research fields would support the use of special experimental devices, such as mobile data collectors and sensors located in hostile areas, without physical or economic difficulties (such as installing cables between a monitoring server in the laboratory and a sensor station located at the top of a mountain). Second, sneakernet refers to the transfer of digital data via physical media such as magnetic tapes, floppy disks, optical discs, flash drives, or even external hard drives. This data collection scheme is quite simple but also time-consuming. Data transfer through wireless links would reduce the time required for data collection and enhance the efficiency of research activities. Finally, cable salad refers to untangling knots or the entanglements of cables. Repetitive experiments involving the relocation of experimental equipment, rewiring, and the movement of wired devices might result in unintentional cable salad. This phenomenon disrupts future experiments and exhausts researchers, as it necessitates the recovery of a well-arranged research environment.

In this section, we briefly review the wireless communication technologies used for the IoT and introduce KREONET, a representative research and education network in Korea. Among the various wireless technologies, we mainly focus on LPWAN technology, which enables scientific data transmission even under hostile conditions owing to its wide coverage and robustness. By contrast, KREONET provides high-performance, dedicated network services to scientists and researchers. Owing to advances in proliferation of the data-oriented research paradigm, KREONET is aiming to build wireless IoT infrastructures for accommodating the requirements of various research areas such as environment, energy, agriculture, forestry, and oceanology.

### 2.1. Wireless Communication Technology for IoT

Given that the IoT is being adopted in various applications, many wireless technologies are being employed considering their performance and features. According to their wireless coverage and data rate, wireless technologies can be categorized into wireless wide area networks (WWANs), wireless local area networks (WLANs), wireless personal area networks (WPANs), and LPWANs as shown in Figure 2. WWANs, WLANs, and WPANs are suitable for mobile device communication, building/office networks, and home/industrial dedicated networks, respectively. However, we focus on LPWAN technologies, which are suitable for collecting data from sensors located over distributed and possibly hostile areas. This is required in various research fields, including environmental, energy, and agricultural research.

Table 1 shows a comparison of the three representative LPWAN technologies: Narrowband IoT (NB-IoT), SigFox, and LoRaWAN. Many studies have been conducted to investigate their similarity and the distinctive characteristics among them [3,4,5,6]. NB-IoT uses licensed frequency bands standardized by 3GPP and provides long-range communication and a high data rate. Therefore, it incurs high costs due to the expense of dedicated communication frequency bands and requires a considerable amount of battery power. By contrast, SigFox and LoRaWAN are operated in the unlicensed Industry-Science-Medical (ISM) frequency bands, thereby circumventing the cost for dedicated communication spectra. Although these technologies provide a lower data rate than cellular technology, they support very long-range communication with low energy consumption. Among the unlicensed LPWAN technologies, LoRaWAN offers advantages in terms of security, adaptability, and privacy. Thus, ScienceLoRa was built based on the LoRaWAN technology to provide data transmission in research and education.

LoRa/LoRaWAN is composed of a physical layer (LoRa) and a network layer (LoRaWAN). The protocol stack of LoRa/LoRaWAN comprises the physical, media access control (MAC), and application layers. LoRa is the physical layer of LoRaWAN; it is responsible for establishing long-range communication links based on proprietary modulation technology that operates in the sub-GHz ISM frequency band. For example, it operates at 920–928 MHz in the United States, at 863–870 MHz in Europe, and 920–923 MHz in Korea. LoRa adopts the chirp spread spectrum (CSS) modulation scheme, which is robust against external interruptions such as channel noise and multipath fading. Therefore, it is suitable for low-power and long-range wireless communication in various IoT environments. CSS uses linear modulated chirp pulses to encode information. A chirped pulse is a sinusoidal signal, the frequency of which increases or decreases over time. LoRaWAN specification for the recommendation of regional parameters [7] defines several classes of data rate (DR) that consist of the corresponding spreading factors (SF) and bit rates (bit/sec) according to the regional regulations. A higher SF results not only in better sensitivity (i.e., the under bound of received signal strength indicator (RSSI) for successful demodulation) with a higher time-bandwidth product but also in a longer time-on-air (ToA) owing to the low bit rate. The signals modulated by different SFs are quasi-orthogonal; hence, a LoRa gateway can receive multiple signals with different SFs simultaneously, through which the low data rate can be supplemented sufficiently. The robustness of the LoRa is suitable for collecting scientific data from desolate or even dangerous areas, without any intervention from scientists.

LoRaWAN is an open specification managed by LoRa Alliance [8,9] and operated in the upper layers of LoRa [10]. A LoRaWAN network consists of four main components: end node, gateway, network server, and application server. The end nodes, such as sensors, collect data, and these collected data are transmitted to the gateways. The gateways relay the uplink/downlink packets transparently between the end nodes and the network server through back-haul networks such as cellular networks, Ethernet, and Wi-Fi. The network server is responsible for supervising the network topology, managing the uplink/downlink traffic status, and configuring the communication parameters of the end nodes and gateways. Thus, the network server ensures reliable and secure data transmission along with the network. For data security, the data between end nodes and the network server are encrypted under AES 128 (Advanced Encryption Standard 128) and transmitted. The application servers obtain data from the network server and provide the services for users.

The end nodes are classified into Class A/B/C depending on the reception mode for accepting downlink packets. A Class A node performs an ALOHA-based uplink packet transmission, which is immediately followed by two short receiving windows. The autonomous uplink packet transmission followed by two short receiving windows is the basic operation that needs to be supported by Class B and C nodes as well. Although Class B or Class C nodes are suitable for reliable and real-time downlink reception, they are not widely used owing to the energy consumption problem caused by the periodic or continuous activation of the communication module for receiving downlink signals. Therefore, Class A is the dominant type in various applications such as remote meter reading and environment monitoring. Detailed specification and their potential performance can be found in LoRa/LoRaWAN standards [7,10] and many mathematical and experimental studies [11,12,13].

### 2.2. KREONET: Korea Research Environment Open Network

KREONET is the representative research and education network in Korea. The goal of KREONET is to provide a collaborative research environment for a variety of research areas such as high-energy physics, astronomy and space observation, genomics and biology, weather/climate, education, future networking, and supercomputing. Since 1988, KREONET has been providing networking services, and it is currently focusing on realizing a science big data super highway through ScienceDMZ, ScienceLoRa, and SDN (software-defined network). More than 200 organizations in Korea, including universities, research centers, and government institutes, are connected through KREONET. KREONET currently operates and manages 17 regional centers in Korea and 4 international Points of Presence (PoPs) (i.e., Hong Kong, Seattle, Chicago, and Amsterdam). Particularly for international networking services, KREONET has been participating in the GLObal RIng network for Advanced Application Development (GLORIAD) project since 2005. Figure 3 and Figure 4 show the network topologies of KREONET and GLORIAD, respectively.

## 3. ScienceLoRa: IoT Infrastructure for Science Research and Education

ScienceLoRa is one of the revolutionary projects of KREONET for expanding the supporting research areas and penetrating network coverage. In 2018, KREONET constructed ScienceLoRa, which is a wireless IoT infrastructure based on LoRa/LoRaWAN technology for scientific research. Along with the characteristics of LPWAN, ScienceLoRa is especially suitable for research areas involving low-rate data collection from many sensors over a wide area, such as in agriculture, environmentology, and oceanography. Moreover, the research experiments conducted in hostile or mobile environments, such as underground research facilities and experiments on autonomous vehicles, can be supported by ScienceLoRa. ScienceLoRa is an access network that provides a wireless connection to researchers for collecting a variety of data. The data collected by ScienceLoRa are transferred through KREONET and shared with researchers worldwide.

### 3.1. Network Architecture

Figure 5 shows the ScienceLoRa network architecture consisting of ScienceLoRa-Device, ScienceLoRa-Gateway, and ScienceLoRa-Network Server. Users such as engineers and researchers can use the ScienceLoRa infrastructure through the ScienceLoRa-Network Server by attaching their application servers. A ScienceLoRa-Device includes sensors and a LoRa communication module; these are connected by a serial interface. A ScienceLoRa-Device collects sensor data and transmits the data to a ScienceLoRa-Gateway. These ScienceLoRa-Gateways send packets to a ScienceLoRa-Network Server through KREONET. Finally, the packets are sent to the users through database servers, web servers, and user terminals. For deploying the ScienceLoRa network, the ScienceLoRa-Device and ScienceLoRa-Gateway were developed in a programmable manner, and the ScienceLoRa-Network Server is installed with high availability (HA) via duplication.

### 3.2. Hardware for Device and Gateway

The ScienceLoRa-Device was developed for accommodating both SX1276 (operated in the sub-GHz ISM bands for low data rate long-range links) and SX1280 (operated in the 2.4 GHz ISM bands for high data rate mid-range links) transceivers. Compared to the sub-GHz LoRa, which has been intensively investigated [15,16], the 2.4 GHz LoRa has only been developed recently for providing global availability and improving data rate with a wider bandwidth and higher frequency under the same duty cycle regulation [17]. The device consists of two parts: a LoRa module for transmission and the main body, which is a small-sized machine with Linux. The Linux machine provides serial interfaces for receiving data from sensors. The collected sensor data are transmitted through the LoRa module. If the packet transmission is not performed appropriately, the un-transmitted packets are stored in the device and retransmitted once the system returns to its normal status. Figure 6a shows an example of the ScienceLoRa-Device developed for an environmental radiation monitoring system (see Section 4.1).

A ScienceLoRa-Gateway comprises four USB-type interfaces for connecting LoRa transceivers. A LoRa communication module (i.e., SX1276 or SX1280) can be plugged into the USB-type interface in the ScienceLoRa-Gateway. The ScienceLoRa-Gateway can be connected via IEEE 802.3 and 802.11 n/ac to send packets to KREONET. Figure 6b shows the developed ScienceLoRa-Gateway. The ScienceLoRa-Device and ScienceLoRa-Gateway in Figure 6 are custom-made hardware platforms for accommodating the special requirements in terms of size, interface, security, and connectivity from the users. However, the off-the-shelf LoRa devices can be integrated into ScienceLoRa because the ScienceLoRa-Network Server is built based on the LoRa/LoRaWAN standard.

### 3.3. Network Protocol and Web-Based Management

End-to-end communication (i.e., between a ScienceLoRa-Device and an external user) is divided into two parts: data collection through LoRa/LoRaWAN and data distribution; this is shown in Figure 7. The data collection part is based on the LoRa/LoRaWAN standards to ensure that ScienceLoRa might accommodate other standardized end devices and gateways. Moreover, ScienceLoRa provides programmable interfaces for ScienceLoRa-Device and ScienceLoRa-Gateway. Accordingly, users can customize networking parameters such as bandwidth, channel, and modulation/encoding schemes to enhance their network performance or to investigate wireless networks. By contrast, the data distribution part is composed of TCP/IP connections and the message queuing telemetry transport (MQTT) protocol. Users can register their gateways and devices through the web interface and monitor real-time data through MQTT.

As mentioned above, the data distribution part is further divided into two subparts. First, ScienceLoRa exploits the MQTT protocol for real-time data monitoring. MQTT is commonly used for transporting small IoT data and it provides publish-and-subscribe messaging based on a hierarchy of topics. In ScienceLoRa, the topic contains the user ID and MAC address of the device to ensure the correct end-to-end connection. Second, ScienceLoRa provides registration, management, and data sharing through the web interface. In addition to the network server, ScienceLoRa includes a database and a web server for management and data sharing services. The web server allows for the registration and management of users, registration of gateways and end devices, condition monitoring of the devices, real-time data monitoring, and configuration of QoS policies. Moreover, the historical data (i.e., not real-time but previously accumulated data) can be provided to the user via the web interface. To this end, the database includes management information, such as user ID, GW ID, sensor ID, and health information of the devices, and also historical data, including the type and location of the sensors, sensor data value, and sensor data generation time. Figure 8 shows the web-based user interface of the (a) ScienceLoRa-Network server and (b) ScienceLoRa-Gateway.

In terms of network management, as shown in Figure 7, there are two types of communication links: Secure SHell (SSH) links and LoRaWAN control links (i.e., through LoRaWAN MAC Command). To directly manage the web server, database server, networks server, and gateways, the SSH tunnels are established between the administrator and each entity. Through the management links, the administrator can check hardware status, update network configuration, manipulate collected data, and collect operational logs. On the other hand, the logical LoRaWAN control links are required for managing ScienceLoRa-Devices because a direct TCP/IP link cannot be established. Along the control links, the administrator and a LoRaWAN device exchange LoRaWAN MAC commands. The hardware status and communication parameters (i.e., DR, SF, uplink/downlink channels, and security) can be configured with the control links.

## 4. Current Services of ScienceLoRa

ScienceLoRa can be used for collecting research data not only over a wide area but also from hostile or isolated environments using long-distance communication links. That is, the high link budget and robustness of ScienceLoRa can overcome a poor wireless environment (e.g., non-line-of-sight paths or severe multipath fading) and the difficulty of building wired infrastructure. Moreover, even in research areas where commercial network infrastructure is available, ScienceLoRa assists the researchers by providing a secure and programmable network that they can use to build a customized infrastructure. In addition to these characteristics, three innovative research services are provided: environmental radiation monitoring, underground research facility safety, and campus IoT research service. Figure 9 shows the current services of ScienceLoRa and its system deployment. Currently, 56 ScienceLoRa-Devices and 25 ScienceLoRa-Gateways are deployed for five different research projects. The research projects include an environmental monitoring and safety system managed by national research institutes and a few lab-scale experiments at universities.

### 4.1. Environmental Radiation Monitoring Service

The Korea Institute of Nuclear Safety (KINS) has been operated and managed the Integrated Environmental Radiation Monitoring Network (IERNet), which is an automated surveillance network for efficiently monitoring environmental radiation in Korea. Figure 10 shows that IERNet consists of 171 surveillance sites and environmental radiation detectors located at each site. The data collected from the radiation detectors are transmitted to the central server at KINS and provided to the public in real-time.

As shown in Figure 11a, IERNet transmits the collected data using a commercial 2G mobile network. The data collected from the radiation detectors are transmitted through the 2G cellular network to a base station. Finally, these data arrive at the application server of the central system at KINS. The transmission architecture based on the mobile network offers an advantage in terms of management and operation overheads. However, two significant problems arise when conducting environmental radiation monitoring through 2G mobile networks. First, important environmental radiation data need to be transmitted through the networks of commercial mobile service providers. The security and integrity of these environmental radiation data should be guaranteed given that they are scientific data; however, one cannot guarantee the safe and reliable data transmission of these environmental radiation data. Second, IERNet is facing stability problems such as in the quality of service due to aging 2G network infrastructures. In addition, the mobile network operators plan to shut down 2G cellular network services due to the reduced 2G users and the increased operational costs for managing superannuated 2G network equipment. Because of these issues, the safe and reliable data transmission of environmental radiation data cannot be guaranteed.

To overcome the above-mentioned problems, we utilized secure data transmission and independent establishment of the network through ScienceLoRa. Figure 11b shows the transmission architecture of the IERNet using ScienceLoRa. The collected data are transmitted by the ScienceLoRa-Devices connected with the detector to ScienceLoRa-Gateways. The data finally reach the application server of the central system at KINS via the ScienceLoRa-Gateways and ScienceLoRa-Network servers. Thus, ScienceLoRa enables the reliable and safe transmission of environmental radiation data, because the data are transmitted through KREONET. Furthermore, the IERNet can provide stable service without depending on the mobile network providers. In the ScienceLoRa-Gateways for IERNet, SX1280 transceivers using the 2.4 GHz band were employed, and a speed of approximately 15 kbps was achieved while transmitting from the ScienceLoRa-Device to the IERNet-Service Server. Furthermore, for reliable and secure packet transmission, the ARIA (block cipher algorithm for Academy, Research Institute, Agency) algorithm is used to encrypt packets between the ScienceLoRa-Device and the IERNet-Service Server.

ScienceLoRa-Gateways have been installed on 19 selected environmental radiation monitoring sites. The left side of Figure 9 shows the deployment sites for the ScienceLoRa-Gateways. As a demonstration project, six monitoring sites were installed at Daejeon in 2018, which include Daedong, Yuseong, OEMC, Dong-gu, Gwanpyeong, and Daedeok. The first phase of the expansion project was launched in 2019, and 13 additional monitoring sites were installed nationwide, including Sanyang, Deokpung, Sejong, Habuk, Geumgansong, Gwanghoe, Haeri, Sindongho, Buan, Gangnam, Mara, Yeosu, and Ulsan.

### 4.2. Safety Service in Underground Research Facility

Globally, studies based on underground research facilities are actively conducted to search for dark matter and investigate the characteristics of neutrinos. Representative underground research facilities include SNOLAB (Canada) [18], LNGS (Italy) [19], and SURF (USA) [20]. The center for underground physics of the Korea Institute of Basic Science (IBS) [21] is also conducting research activities such as nuclear and particle experiments, searching for dark matter/neutrinos, and developing low-temperature detection technology at Yangyang (Y2L) and Jeongseon (Yemilab). Y2L has two laboratories including COSINE and AMoRe experimental equipment, and the total area is 300 m${}^{2}$. Figure 12a shows the layout and organization, while Figure 12b shows the supporting infrastructure and experimental equipment in Y2L. The other facility, Yemilab, located at a depth of 1km from the summit of Yemi mountain, is an underground research facility aimed at searching for dark matter. It consists of twenty-six laboratories for eight experiments and ten support rooms. Figure 13a shows a 3D image of Yemi mountain, while Figure 13b shows the layout and organization of Yemilab. Each research facility has supporting infrastructures for supplying electricity and managing air conditions.

In an underground research facility, the safety monitoring systems are important to ensure the safety of researchers and to control the laboratory environment (i.e., temperature, humidity, and air quality) for preventing accidents. Additionally, visitors’ locations are not monitored; instead, a simple confirmation of access authorization at the entrance is conducted. Therefore, the information about whether a visitor is in the facility or not is only available. Additionally, the environmental monitoring of the underground facilities might be integrated with the safety systems to ensure the awareness of dangerous situations such as a lack of oxygen, poor air quality, and the leakage of harmful gases. In this context, for the appropriate lifesaving measures in emergencies, location and physical condition monitoring and environmental monitoring without human intervention are essential.

To provide IoT-based safety services in underground research facilities, ScienceLoRa was deployed in Y2L and Yemilab. Through the ScienceLoRa network, the real-time location was monitored by indoor localization. In addition, the researchers can collect a variety of environmental sensing data such as temperature, humidity, vibration, wind speed, and oxygen level. Figure 14 shows the operation of ScienceLoRa in the underground location and environment monitoring system. The system consists of a ScienceLoRa-Gateway, ScienceLoRa-Devices, wristbands, location beacons, and environmental sensors. The ScienceLoRa-Device is carried by each researcher as the ID card and includes a Bluetooth Low Energy (BLE) device for collecting the surrounding data. The wristbands are also carried by a researcher to generate biometric data, including body temperature and heart rate. The location beacons and environmental sensors generate the location identification and sensor data, respectively. The card-type ScienceLoRa-Device exchanges collected data (i.e., location, bio-metric, and environment) and control messages with the ScienceLoRa-Gateway through the LoRa interface. The top right images in Figure 9 show the deployed ScienceLoRa in the underground research facility.

### 4.3. Campus IoT Service for Research and Education

ScienceLoRa provides network infrastructure not only for the IoT R&D in science applications but also for educational purposes for students. Currently, ScienceLoRa is collecting environmental data such as temperature, humidity, fine dust, harmful gas, and solar radiation, which are widely used for agriculture, energy, and environmental studies in cooperation with universities. There are three use cases of the ScienceLoRa-based campus IoT infrastructure. First, researchers can directly use data for their research projects. Second, researchers can build any customized research system, such as research facility monitoring, research environment management, and event alarm systems. Third, IT courses can use the ScienceLoRa infrastructure to develop devices in an embedded system class, analyze big environmental data in data science classes, and configure communication parameters in computer communication classes. The bottom right images in Figure 9 show the deployed ScienceLoRa in campuses including KISTI, Chonnam National University, and Kumoh National Institute of Technology.

Researchers can build their services based on ScienceLoRa. Figure 15 shows two typical use cases. First, the real-time monitoring service visualizes the collected data so that researchers can observe real-time information. This type of service may be used in research facility monitoring and environmental monitoring. Second, the rule-based alarm service sends urgent messages to the user’s smartphone, email, and web application when a critical event occurs. Researchers might use this alarm service in accident detection (e.g., fire, gas leakage, explosion) and invasion detection (e.g., sensitive research facilities and security equipment). Furthermore, we provide a data-sharing service for exchanging multi-sourced research data among researchers for developing cooperative research projects. The data-sharing system was developed based on Nextcloud [22], which is open-source client-server software that provides secure and convenient data file sharing, as shown in Figure 16.

## 5. Evolution of ScienceLoRa towards ScienceIoT

Recently, along with the data-centric research paradigm, the demand for collecting multimedia data (e.g., images and videos) and using mobile/distributed IoT devices (e.g., UAVs and sensors) has been growing. Application scenarios include collecting image data by using drones, preventing safety accidents based on mobile robots, and gathering environmental data from distributed sensors. Moreover, wireless connectivity among research facilities through the standardized protocols in license-free bands would enhance the efficiency of research activities in terms of cost and interoperability.

ScienceLoRa is an LPWAN-based IoT infrastructure that supports research and education activities. However, LoRa has a limitation in transferring multimedia data owing to its low data rate. In addition, the simple network architecture of LoRaWAN might cause data concentration on the network server if the network size is growing. Therefore, the current underlying technologies and network architecture of ScienceLoRa should be integrated with various technologies to provide massive/high-speed data communication, thereby supporting future data-centric research activities. As a next step, therefore, ScienceLoRa is evolving into ScienceIoT by accommodating advanced wireless technology and edge computing capabilities to enhance the data rates and mitigate data concentration. Based on ongoing projects, ScienceIoT will expand the coverage of scientific and educational services such as smart research environments and test-beds for wireless communication and network research.

Figure 17 shows the comprehensive vision of ScienceIoT including existing capabilities. The integrated ScienceIoT platform consists of four parts: edge management, web-based management, data sharing, and computing infrastructure. In edge management, the edge nodes (residing on gateways, access points, and base stations) are configured, and the collected data are pre-processed. Web-based management deals with the registration of devices, device state monitoring, the configuration of hardware/network settings, and firmware updates. Finally, the users of ScienceIoT can obtain the collected data through a file-sharing system or a real-time monitoring protocol. In addition, the collected data can be delivered to a DTN of ScienceDMZ (see Section 2.2) or the supercomputing infrastructure of KISTI. The parts that are most different from previous ScienceLoRa architectures are diversified wireless access networks and the edge computing structure. First, ScienceIoT provides high-speed and massive scientific data collection services by embracing 5G and Wi-Fi networks. The expanded wireless capability enables local-scale or lab-scale high-rate data collection as well as current wide-scale low-rate data collection. Second, the edge node that resides in the gateway/access point/base station of each access network provides edge computing capabilities. Concretely, they perform pre-processing on the collected data, de-duplication, error recovery, and data integration. The edge functionalities reduce the overheads of the backbone network and offer the opportunity for local intelligence.

### 5.1. Accommodating High Performance Access Network

Recently, research projects based on wireless communication have been actively conducted in the fields of sensor data collection, multimedia data transfer from UAVs, and remote robot control. For example, a research project on smart agriculture is collecting visual data using drones, where 1.55 TB of image data is captured during a flight. Current data transfer from the drone to the research facility takes several hours, including transportation from the data generation field to the laboratory and disk-to-disk copy. To reduce the data collection delay, ScienceIoT has to embrace wireless technologies providing high-speed data transfer. As representative candidates, we introduce two access network technologies: 5G and Wi-Fi.

As 5G mobile network technology has rapidly developed, high-performance wireless communication has become available. Although private 5G is mainly implicated in the industrial field as discussed in [23], the advantages of 5G technology such as dedicated coverage, customized service, and dependable communication are commonly demanded by scientific research projects. In addition, a high-level communication performance is also required in research networks due to the data-oriented research paradigm. In this sense, 5G networks provide a very high data rate of up to 20 Gbps and a very low latency of at least 1 ms. Along with the high-performance, the mobile networks are suitable for supporting mobile research devices such as drones monitoring areas of interest and automobiles containing various mobile sensors. To exploit the advantages of 5G, global NRENs are carrying forward various plans for adopting 5G technology into their research networks in terms of technology and policy. In addition, in many countries, some parts of 5G frequency bands are defined to be used for industrial or public purposes. This separation would encourage the research field to use private 5G networks without high operational costs. A private 5G network might be dependently or independently constructed to the commercial 5G network. There are many variants according to which component (e.g., base station, edge server, or core network system) is to be shared between private and commercial networks. From the view of research networks, sharing more components would reduce the capital and operational costs, whereas independent private networks would provide the authority of network configuration and higher security.

Wi-Fi is a wireless technology based on non-commercial frequency bands. Therefore, NRENs usually exploit Wi-Fi for providing campus-wide Internet connectivity. The most powerful aspect of Wi-Fi is the extremely high availability. In other words, installing a Wi-Fi access point makes a wireless network. The scaling up is easily achieved by binding multiple access points into a LAN. Therefore researchers can build a wireless network in their laboratory or research facility without expertise in networking. In addition, as Wi-Fi has been improved from 802.11a to 802.11ax (Wi-Fi 6), its data rate and reliability have increased dramatically. Moreover, 802.11be (Wi-Fi 7), which is a work-in-progress amendment by the Extremely High Throughput (EHT) task group, is expected to provide a maximum throughput of at least 30 Gbps [24]. The main limitation of Wi-Fi is that it does not provide determinism because of the multiple access mechanisms based on contention avoidance. Therefore, it should not be used in situations where some unexpected delay causes a fatal accident. Nevertheless, researchers can use Wi-Fi as an easy-to-use and high-performance wireless network for monitoring laboratory/research facilities and connecting their wireless devices.

The research for accommodating high-performance wireless technologies includes designing an integrated network architecture based on license-free communication protocols, devising network slicing techniques for high-performance data transmission, and interconnecting a high-speed wired backbone (i.e., ScienceDMZ) [25] and wireless networks. In addition, flow management and communication mechanisms should be developed according to the application scenarios. For example, the 5G use cases consist of enhanced mobile broadband for high-speed data transmission, massive machine-type communication for accommodating a large number of end devices, and ultra-reliable and low latency communication to support mission-critical applications. Furthermore, the special requirements of each research field such as privacy, QoS, and multi-level security would be supported by the integrated wireless network.

### 5.2. Smart Edge for Distributed IoT Data Processing

Through KREONET, multi-sourced scientific data have been delivered to multiple researchers. As the amount of scientific data is continuously growing, the overhead of the wired backbone network increases in terms of traffic congestion and the corresponding delay. As mentioned at Section 1, when the number of data sources increases, current centralized network architecture causes network congestion. In addition, the use of multimedia data in the scientific field aggravates this situation. In this context, edge computing technology should be applied to reduce the overhead of the backbone network and to effectively conduct the data collection and process the data. In other words, moving computation capability from a centralized cloud to the edge of the network (e.g., gateways, base stations, border routers, and so on) leads to effective data collection and processing for resource-limited IoT devices [26].

As an ongoing project toward ScienceIoT, we are developing an edge computing system based on EdgeX Foundry [27], an open-source edge computing platform. EdgeX Foundry is designed to ingest data from multiple sources and then forward those data to a central system. EdgeX Foundry is composed of a collection of microservices, thereby providing flexibility in terms of service integration and interoperability. The interfaces for devices (i.e., the communication/network/application protocols), key functions (e.g., rule engine, scheduling, and logging), and additional services can be implemented as a microservice and integrated into the EdgeX Foundry platform. Therefore, the user can customize the platform by adding, removing, and modifying the microservices.

From the view of ScienceLoRa, the EdgeX Foundry platform affects the efficiency of the data collection and data distribution. ScienceLoRa-Devices would be implemented as one of the device services, and EdgeX Foundry would be installed on the ScienceLoRa-Gateway or ScienceLoRa-Network Server. In this context, EdgeX Foundry reduces the overhead of the wired backbone network by filtering, aggregating, and pre-processing data. For example, in the environmental monitoring service, duplicated or erroneous data can be filtered while multi-dimensional data from various sensors can be integrated into a formatted tuple. In addition, the future ScienceIoT architecture would include various wireless technologies, such as private 5G, WLAN, and Bluetooth. Based on high-performance wireless communication, an edge node might conduct computation-intensive tasks. For example, in a visual monitoring service where a drone collects image or video data from an interesting area, an edge node can perform object recognition or classification based on a deep neural network. The EdgeX Foundry instances installed at the gateways/access points of each access network can perform formatting and aligning to integrate various types of devices and protocols.

The research for building IoT edge computing infrastructure based on a smart-edge platform for ScienceIoT includes extending license-free IoT network infrastructure, designing an efficient edge network architecture, and developing a smart-edge core architecture for efficient scientific data processing.

## 6. Conclusions

In this article, we introduce the underlying technologies and scientific services of ScienceLoRa and present the future direction of ScienceIoT. ScienceLoRa is a wireless IoT network infrastructure that can be used for various scientific applications by collecting research data based on the LoRa/LoRaWAN technology. ScienceLoRa is a dedicated network for research and education based on KREONET, which enables reliable and secure data transmission. Currently, ScienceLoRa is used for environmental radiation monitoring, safety systems in underground facilities, and research/education networks at university sites nationwide. In addition, we are designing integrated ScienceIoT architecture for accommodating high-performance wireless technologies and exploiting edge computing capability to provide better data transmission and distributed data processing. Thus, ScienceIoT will support massive data collection and flexible collaboration among researchers who are the driving force toward data-oriented research.

## Figures and Tables

**Figure 1 sensors-21-05852-f001:**
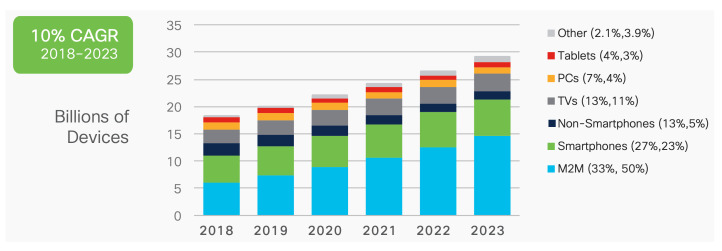
Growth of global devices and connections.

**Figure 2 sensors-21-05852-f002:**
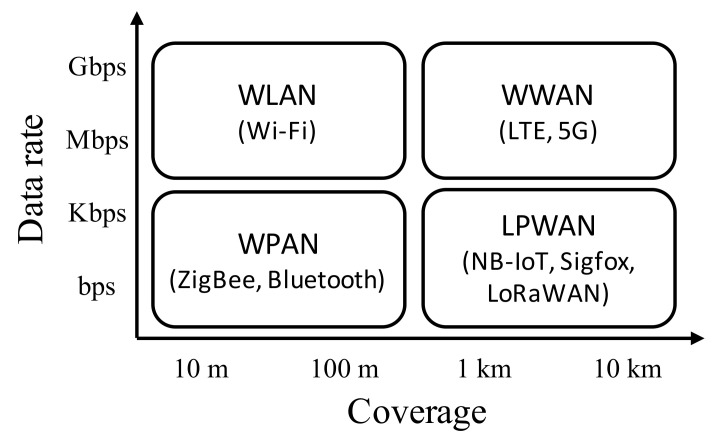
Wireless technologies categorized according to their data rate and coverage.

**Figure 3 sensors-21-05852-f003:**
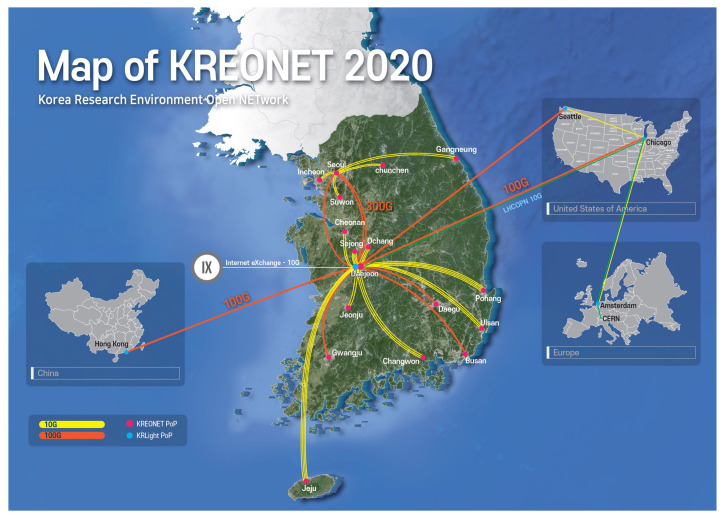
Network map of KREONET, comprising 17 regional centers nationwide and 4 international PoPs worldwide [14].

**Figure 4 sensors-21-05852-f004:**
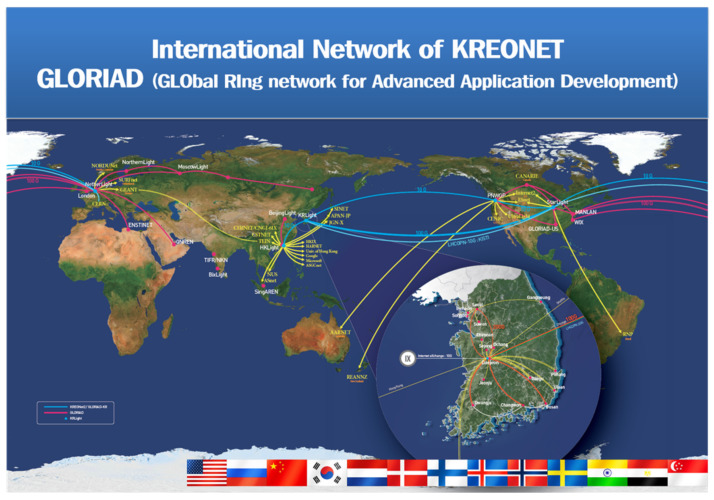
International network map of GLORIAD [14].

**Figure 5 sensors-21-05852-f005:**
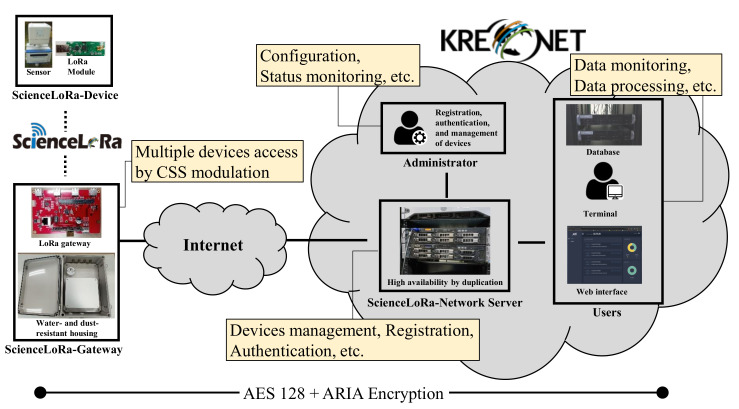
ScienceLoRa network consisting of end device, gateway, network server, and users.

**Figure 6 sensors-21-05852-f006:**
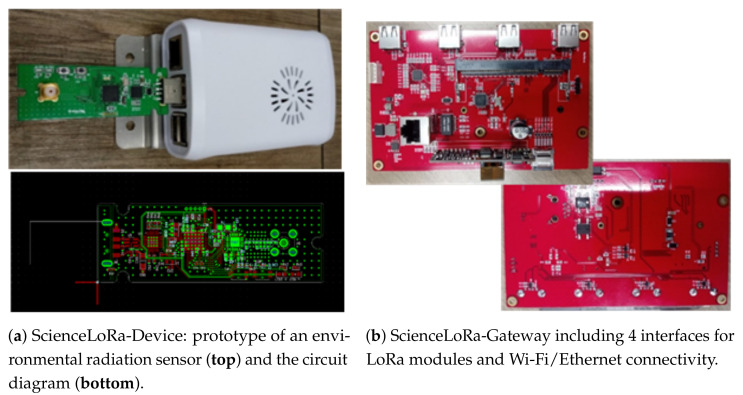
The hardware platforms for ScienceLoRa.

**Figure 7 sensors-21-05852-f007:**
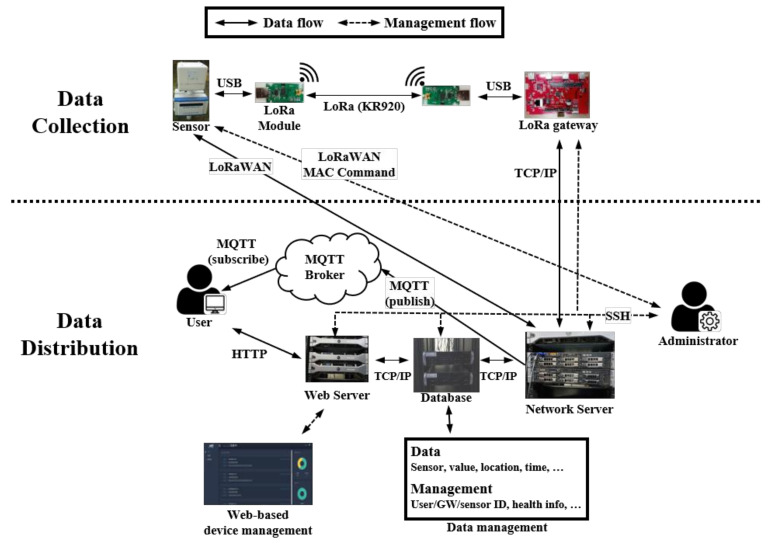
Network and management protocols for ScienceLoRa.

**Figure 8 sensors-21-05852-f008:**
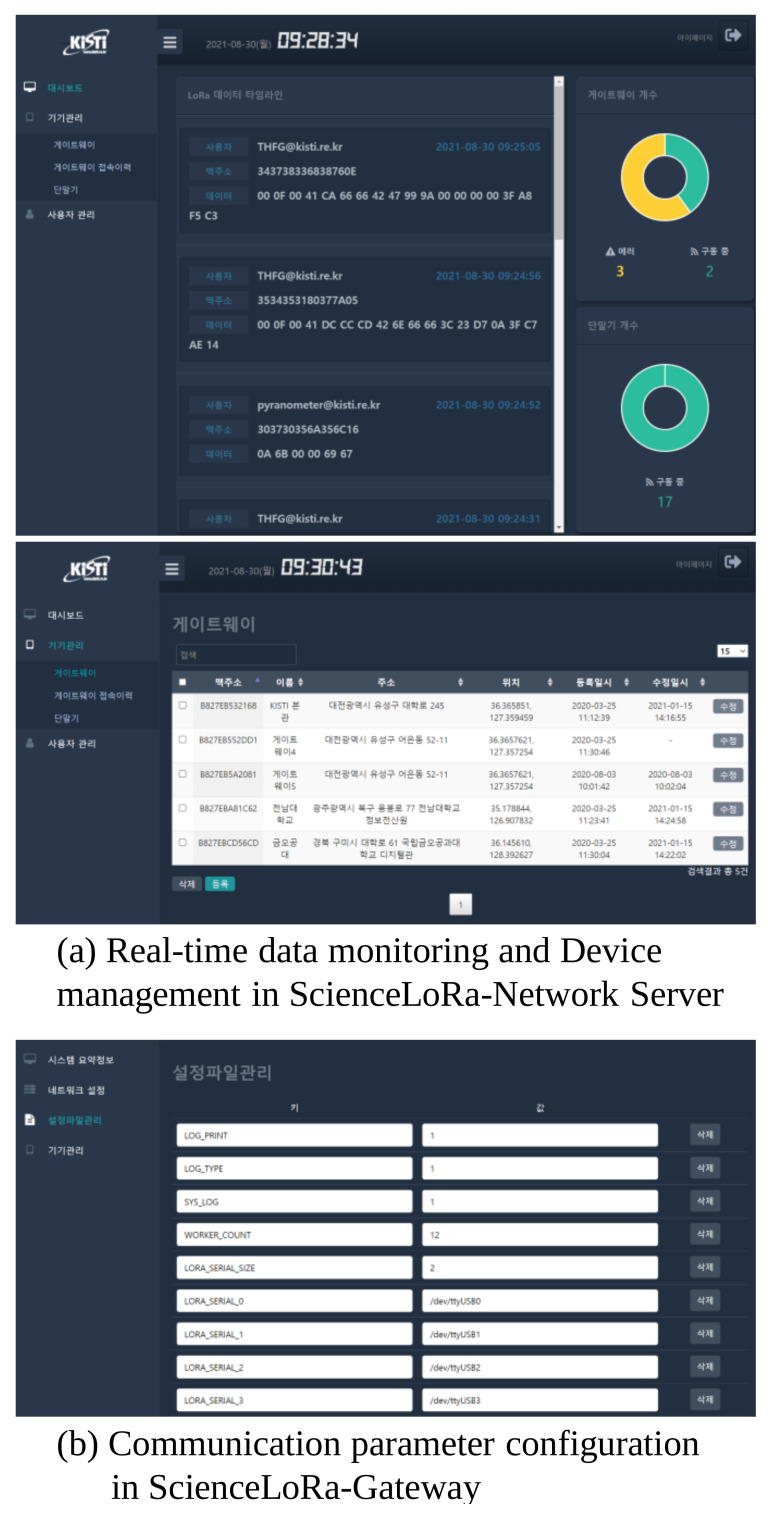
Web-based management system of ScienceLoRa.

**Figure 9 sensors-21-05852-f009:**
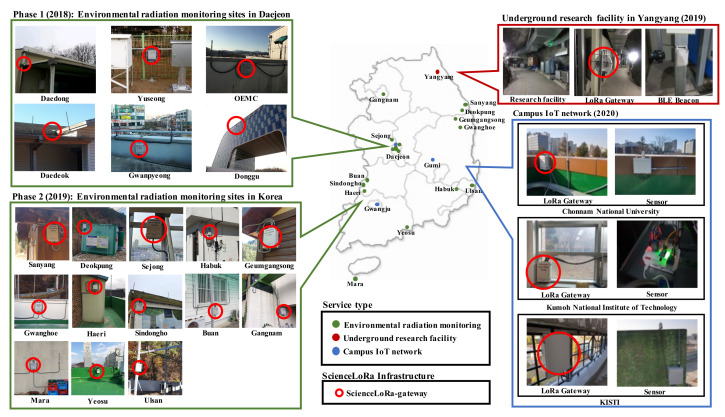
ScienceLoRa Service and system deployment.

**Figure 10 sensors-21-05852-f010:**
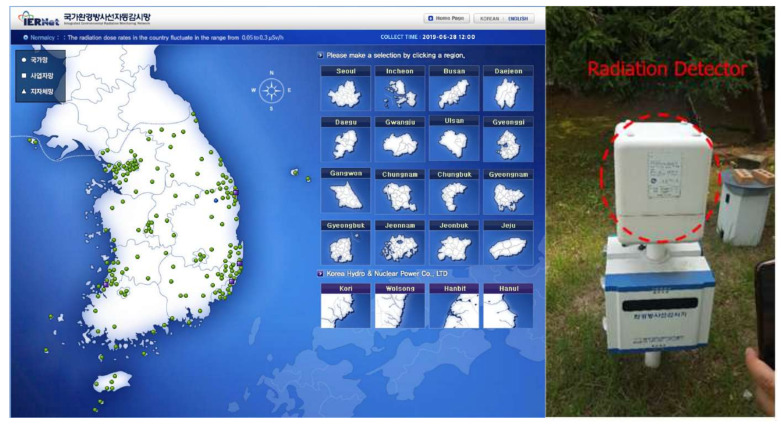
IERNet for environmental radiation monitoring.

**Figure 11 sensors-21-05852-f011:**
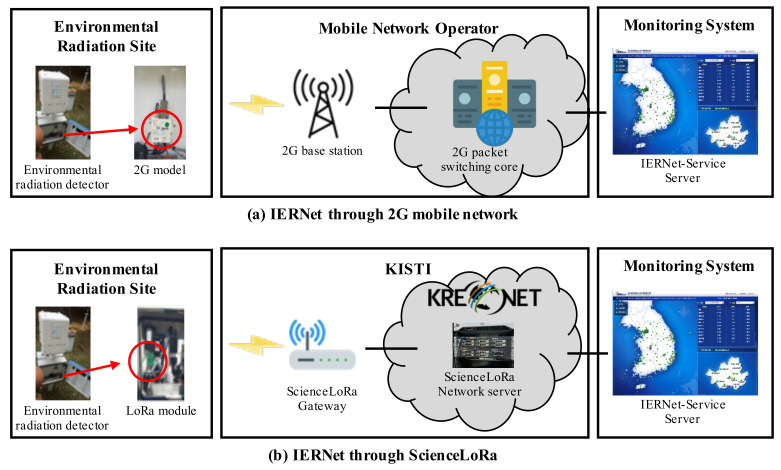
IERNet network architecture through (**a**) 2G mobile network and (**b**) ScienceLoRa.

**Figure 12 sensors-21-05852-f012:**
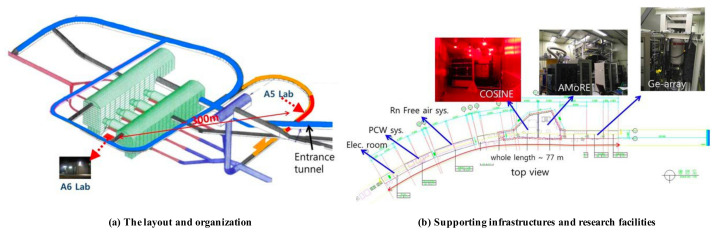
YangYang Y2L. (**a**) The layout and organization and (**b**) supporting infrastructures and experimental equipment [21].

**Figure 13 sensors-21-05852-f013:**
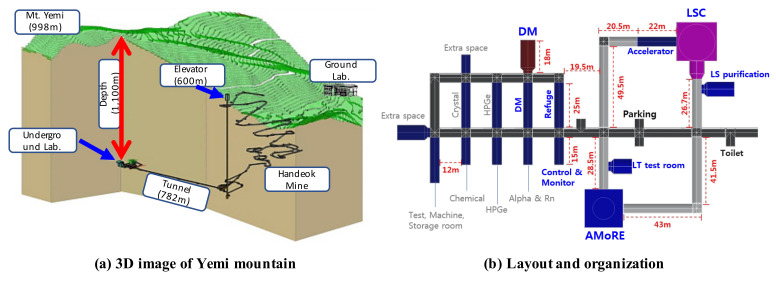
Jeongseon Yemilab. (**a**) Three-dimensional image of Yemi mountain and (**b**) layout and organization of Yemilab [21].

**Figure 14 sensors-21-05852-f014:**
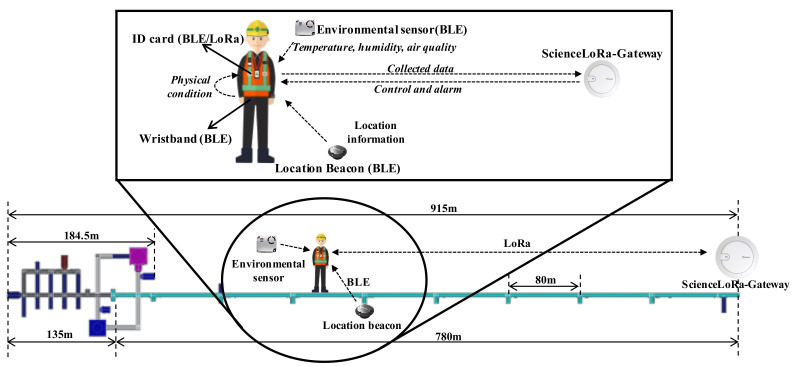
Schematic diagram of the safety service in underground research facility.

**Figure 15 sensors-21-05852-f015:**
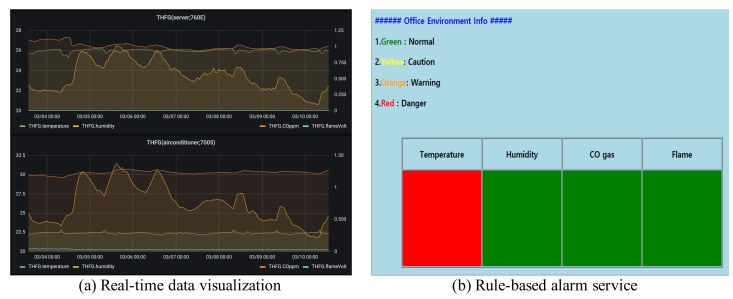
Examples of services using ScienceLoRa data. (**a**) Real-time data monitoring and (**b**) rule-based alarm service.

**Figure 16 sensors-21-05852-f016:**
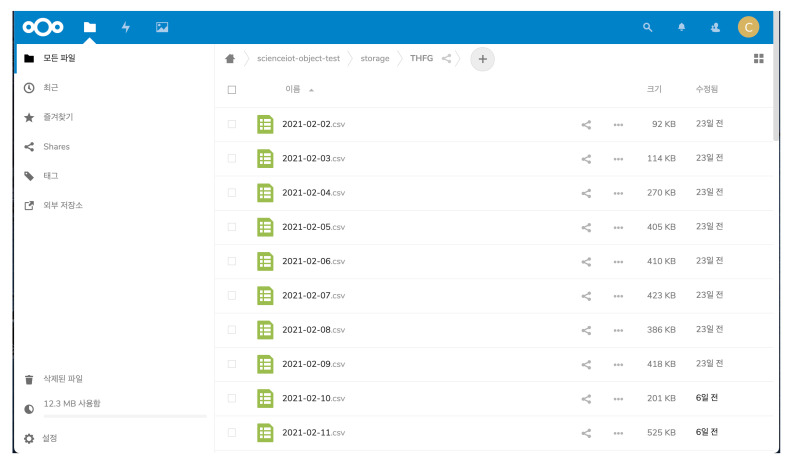
Data-sharing service based on the Nextcloud.

**Figure 17 sensors-21-05852-f017:**
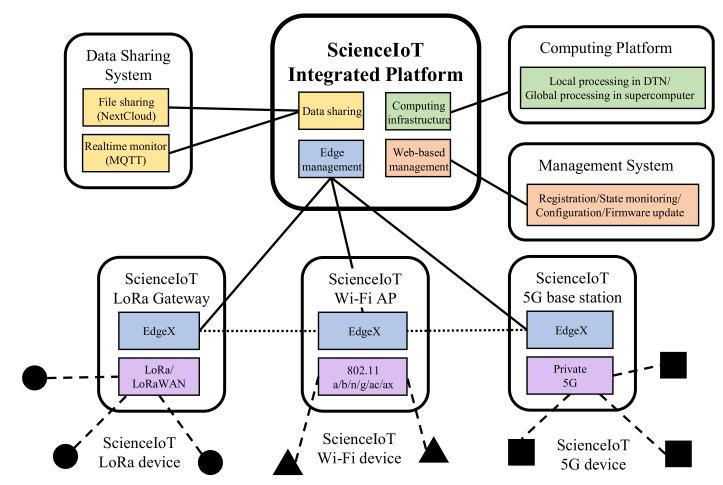
Comprehensive vision of ScienceIoT.

**Table 1 sensors-21-05852-t001:** Comparison of various LPWANs.

	NB-IoT	Sigfox	LoRaWAN
Standardization	3GPP	ETSI and Sigfox Alliance	LoRa Alliance
Technology	LTE (rel.13)	Proprietary	Proprietary
Topology	Star	Star	A Star of stars
Frequency	Licensed LTE frequency band	Unlicensed ISM bands	Unlicensed ISM bands
Duty cycle restriction	No	Yes (typically 1%)	Yes (typically 1%)
Modulation	QPSK/BPSK (DL), GFSK (UL)	DBPSK (UL), GFSK (DL)	CSS/FSK
Bandwidth	200 kHz	100 Hz	125/250/500 kHz
Transmission technique	FDD	UNB	Aloha
Maximum data rate	200 kbps	100 bps	50 kbps
Bidirectional	Yes/Half-duplex	Limited/Half-duplex	Yes/Half-duplex
Maximum messages/day	Unlimited	140 (UL)/4 (DL)	Unlimited
Maximum payload length	1600 bytes	12 bytes (UL)/8 bytes (DL)	243 bytes
Coverage/Range	1 km (urban), 10 km (rural)	10 km (urban), 40 km (rural)	5 km (urban), 20 km (rural)
Maximum TX power	20 dBm/23 dBm	14 dBm/22 dBm	14 dBm/27 dBm
Interference immunity	Low	Very high	Very high
Security	LTE encryption	MAC verification	AES 128
Adaptive data rate	No	No	Yes
Handover	End-device joins a single base station	End-device does not join a single base station	End-device does not join a single base station
Localization	No (under specification)	Yes (RSSI)	Yes (TDOA)
Private network	No	No	Yes
Module cost	<$5	<$5	<$10

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
