# Peer review of "ScienceIoT: Evolution of the Wireless Infrastructure of KREONET"

_sensors, 2021, doi:10.3390/s21175852_

Round 1
Reviewer 1 Report
The work present the concept, current status, applications, and development plans of the wireless infrastructure for science and education in the Korea Research Environment Open NETwork (KREONET). Additionally, ScienceLoRa is evolving toward ScienceIoT by accommodating a private 5G network and an integrated edge computing platform.
The structure of the paper is clear, the language is proper and contributions are well delimited.
Authors should include a related work in order to cite (more scientific papers) and very recent work in this topic and also better explain the novelty of the proposal comparing to the state of the art.
The work is very promising covering a up to date topic, however authors should include and discussing others related initiatives in the topic.
The manuscript needs a revision in order to correct some typos.
Reviewer 2 Report
In this paper, the authors introduce the ongoing efforts for an IoT network for scientific applications in Korea. While the authors provide a wide array of information on IoT systems and some example applications for their platform, specific details on the testbed are lacking. For example, how many nodes are currently deployed? How many users are there for the system? How many users could hypothetically be supported? How many different use-cases are currently present? What are the current bottlenecks of the system? The introduction states that scaling up the number of users will burden the server, but the specifics of the burden are not provided. Does the centralized system become data-limited, IO-limited, or processing limited?
The paper is well written with clear organization. However, the wording is vague in many places. For example, the authors reference "hostile" conditions without clearly stating what is meant. Perhaps the authors mean that there is no line-of-sight path and severy multipath. Or maybe they mean that there is severe interference from other nodes in or outside of the network. The reviewer recommends improving the clarity throughout the manuscript.
There is some repetitiveness throughout that can potentially be streamlined. For example, Figs 7 and 8 are very similar and could be consolidated.
The authors directly use many figures from other sources. In most cases, there is a citation. However, the authors should be sure to appropriately indicated that the figures are not original.
The authors present hardware in section 3.2. The reviewer requests that the authors clearly state if this is a custom device manufactured for this network or a commercial off-the-shelf solution.
There is a significant amount of background information on LoRa. While the high-level background information discussing WPAN around Fig. 2 is overall useful, the low-level details on the LoRa architecture are perhaps less necessary. Interested readers could instead be directed to primary sources on the topic. Most of the details presented are not referenced again and are irrelevant to the novelty of this work.
The transition to a 5G-based system with edge computing provides exciting opportunities for new developments. Like the ScienceLoRa sections, specific details for the system should be provided. The authors describe the edge compute nodes with many generic tasks. Are there any targeted metrics for performance for any particular workload? For example, one widely discussed edge compute workload is deep neural networks for image recognition. Is there a target performance for this or any other workload? Are there any expressions for the number of edge devices needed to process all incoming IoT data? These sorts of details on the exact capabilities could be expanded upon throughout.
The ScienceLoRa and ScienceIoT projects are incredible efforts that have enormous potential. The reviewer hopes that the authors include more details on this work so that the reader can fully appreciate the project and its contributions.
Round 2
Reviewer 2 Report
The reviewer appreciates the efforts to improve the manuscript by the authors and believes there is now more clarity throughout.
One minor point is that there seems to have been a miscommunication about the consolidation of figures. In my original review, I pointed out that Figures 7 and 8 were similar. In the draft I reviewed, this was the "Integrated network architecture ..." and the "Science LoRa network ..." figures which now seem to be Figs 5 and 6. The authors consolidated what I saw as Figs 9 and 10. While the consolidation on the hardware figure was not what the reviewer intended, it is also a welcome modification.
